# Design of AIEgens for near-infrared IIb imaging through structural modulation at molecular and morphological levels

Yuanyuan Li [1,2,6], Zhaochong Cai [3,6], Shunjie Liu [1,2,6], Haoke Zhang [1], Sherman T.H. Wong [1], Jacky W.Y. Lam [1,2], Ryan T.K. Kwok [1,2], Jun Qian [3✉] & Ben Zhong Tang [1,2,4,5✉]

Fluorescence imaging in near-infrared IIb (NIR-IIb, 1500–1700 nm) spectrum holds a great promise for tissue imaging. While few inorganic NIR-IIb fluorescent probes have been reported, their organic counterparts are still rarely developed, possibly due to the shortage of efficient materials with long emission wavelength. Herein, we propose a molecular design philosophy to explore pure organic NIR-IIb fluorophores by manipulation of the effects of twisted intramolecular charge transfer and aggregation-induced emission at the molecular and morphological levels. An organic fluorescent dye emitting up to 1600 nm with a quantum yield of 11.5% in the NIR-II region is developed. NIR-IIb fluorescence imaging of blood vessels and deeply-located intestinal tract of live mice based on organic dyes is achieved with high clarity and enhanced signal-to-background ratio. We hope this study will inspire further development on the evolution of pure organic NIR-IIb dyes for bio-imaging.

[1] Department of Chemistry, Hong Kong Branch of Chinese National Engineering Research Center for Tissue Restoration and Reconstruction, Department of Chemical and Biological Engineering and Division of Life Science, The Hong Kong University of Science and Technology, Clear Water Bay, Kowloon, Hong Kong, China. [2] HKUST-Shenzhen Research Institute, No. 9 Yuexing 1st RD, South Area, Hi-tech Park, Nanshan, Shenzhen 518057, China. [3] State Key Laboratory of Modern Optical Instrumentations, Centre for Optical and Electromagnetic Research, College of Optical Science and Engineering, Zhejiang University, Hangzhou 310058, China. [4] Center for Aggregation-Induced Emission, SCUT-HKUST Joint Research Institute, State Key Laboratory of Luminescent Materials and Devices, South China University of Technology, Guangzhou 510640, China. [5] Ming Wai Lau Centre for Reparative Medicine, Karolinska Institutet, Hong Kong, China. [6]These authors contributed equally: Yuanyuan Li, Zhaochong Cai, Shunjie Liu. ✉email: qianjun@zju.edu.cn; tangbenz@ust.hk

Fluorescence imaging in the second near-infrared region (NIR-II, 1000–1700 nm) enables direct visualization and real-time feedback of deep biological structures with a miraculous degree of clarity than NIR-I (800–900 nm) due to further suppressed photon scattering and minimized autofluorescence[1–7]. Simulation and experimental results have revealed that the imaging performance in term of spatial and temporal resolution and penetration depth could be further enhanced by the NIR-IIb (1500–1700 nm) fluorophores due to almost zero autofluorescence and much lower photo scattering[8–10]. However, fluorophores emitting in the NIR-IIb region are seldom reported. Until recently, remarkable accomplishments have been made by inorganic materials, including quantum dots (QDs)[11–13], rare-earth-doped nanoparticles (RENPs)[14–16], and single-walled carbon nanotubes (SWCNTs)[17], exemplifying the increased resolution for in vivo vessel and tumor imaging. As an alternative, organic materials with the merits of potential biodegradability, salient biocompatibility and ease of processability, hold considerable promise for NIR-IIb imaging[18]. Despite many excellent organic fluorophores emitting at ~1000 nm have been exploited[19–22], fluorescence imaging in the NIR-IIb region is rarely reported mainly due to the shortage of suitable materials with characteristically longer emission wavelength[23].

Extending the conjugation length of organic dyes is a widely explored strategy to redshift the emission. However, when these large π-conjugated systems appear in biologically useful aggregate state or nanoparticles, the strong intermolecular π−π interactions often result in emission quenching[24]. The existence of such aggregation-caused quenching (ACQ) effect makes it difficult to develop bright fluorescent aggregates or nanoparticles for bioimaging. Alternatively, lowering the bandgap of the highest occupied molecular orbital (HOMO) and lowest unoccupied molecular orbital (LUMO) of an organic fluorophore by molecular engineering electron donors (D) and acceptors (A) is another efficient way to redshift the emission. Studies have shown that some fluorophores with distorted D-A architecture exhibit an excited state electron transfer process property, referring to twisted intramolecular charge transfer (TICT)[25–28]. In this process, the emission of the dyes is redshifted in polar solvents such as water but at the cost of fluorescence efficiency owing to the dominated non-radiative decay. If we utilize the merits of the TICT while restricting its non-radiative decay, bright organic fluorophores with emission extended to the NIR-IIb region could be achieved.

On a molecular level, the formation of dark TICT state (a weakly emissive $S_1$ excited state), relies on the flexible intramolecular rotation of D-A units and such motion favors various non-radiative pathways to generate weak but long-wavelength emission (Fig. 1a)[26]. On the other hand, by virtue of restricting of intramolecular motion, enhanced fluorescence intensity can be obtained. As such, fluorophores with both redshifted emission and high quantum yield (QY) can be achieved simultaneously by combining seemingly contradicting individuals, which is a conceptually straightforward but enormous challenge in reality. As a cutting-edge fluorescent technology, aggregation-induced emission (AIE) holds great potential to address this issue[29–33]. AIE luminogens (AIEgens) emit intensely when aggregated because of the restriction of intramolecular motion (RIM) mechanism (Fig. 1b)[34–36]. Noteworthily, thanks to the twisted structures decorated with multiple molecular rotors, AIEgens remain intramolecularly mobile even in the aggregate state, which is inclined to access the dark TICT state[37,38]. Through structural modulation at the molecular (TICT) and morphological levels (aggregation), organic AIEgen-based nanoparticles with long-wavelength emission and high fluorescent QY could be obtained simultaneously (Fig. 1c).

In this contribution, we have designed three D-A typed AIEgens with emission extended to the NIR-IIb region. Strong electron-withdrawing unit benzobisthiadiazole (BBTD) serving as an electron acceptor and triphenylamine (TPA) unit working as both a donor and a molecular rotor was selected to denote the TICT property. In between the BBTD and TPA, alkyl thiophene is introduced to ensure having a large distortion of the conjugated backbone. Notably, 2TT-*o*C26B showed maximum emission at ~1030 nm with a tail extended to 1600 nm and displayed a high QY of 11.5%. As a proof of concept, fluorescence imaging in the NIR-IIb region based on 2TT-*o*C26B organic nanoparticles is reported with high resolution and enhanced signal-to-background ratio (SBR). Importantly, detailed intestinal structures can be clearly visualized in real-time, which can be a powerful "see-through" platform for internal organ imaging.

## Results

**Molecular design.** To construct conjugated AIEgens with bright NIR-IIb emission in the aggregated state, the molecular design has consisted of three elements: (1) a strong D-A structure; (2) rotatable units; and (3) bulky π-conjugation bridges providing certain spatial hindrance to make the molecule have a twisted conformation. BBTD was selected as a strong electron acceptor, whose quinoidal structure admits greater electron delocalization and hence lowers the bandgap (Fig. 2)[39–41]. Meanwhile, alkyl thiophene was applied as the donor unit and π-conjugation bridge. TPA with twisted structure served as the molecular rotor to assure the formation of TICT state while acting as a second donor unit to facilitate the charge transfer. It should be noted that both the alkyl chain's position and the molecular rotors are of key importance for determining the emission of the fluorophores in aggregate. Fluorophores with *ortho*-positioned alkyl chain units (adjacent to BBTD) are associated TPA molecular rotor displaying strong fluorescence in nanoparticles (typical AIE property), owing to the hindrance of strong intermolecular interactions by the twisted structures. It was reported that the alkyl chain could provide spatial isolation of the molecules in nanoparticles to promote intramolecular motions, which was conducive to the formation of dark TICT state[42,43]. In order to investigate the effects of the alkyl chain, linear hexyl unit (2TT-*o*C6B), branched 2-ethylhexyl unit (2TT-*o*C26B), and 2-octyldecyl unit (2TT-*o*C610B) were grafted to the *ortho* position of thiophene. The synthetic routes and structural characterizations of the above compounds can be found in the supporting information (Supplementary Figs. 1–7)[44]. Importantly, key elements of the molecular designs focused on the adoption of second carbon-branched alkyl chains, as they provided tunable steric hindrance not only for preventing intermolecular interactions but also for promoting intramolecular motion[43]. Compared with 2TT-*o*C6B with a linear hexyl unit, 2TT-*o*C26B with branched 2-ethylhexyl unit adopted a larger twisting conformation for intramolecular motion while 2TT-*o*C610B with more hindered 2-octyldecyl unit was expected to have the largest room allowing free intramolecular motions[42]. As shown in Fig. 1 (right), the large dihedral angle (~50°) between thiophene and BBTD confirmed the steric effect of *ortho*-positioned alkyl chains. Moreover, the spatial separation of the HOMO and LUMO was also beneficial for the formation of the TICT state (Supplementary Fig. 8)[45].

**Photophysical properties.** On a molecular level, the three molecules showed typical charge transfer (CT) absorption bands at ~700 nm in tetrahydrofuran (THF) solution[46], while their emission maxima are located at the NIR-II region, providing a platform for fluorescence imaging with a high degree of clarity

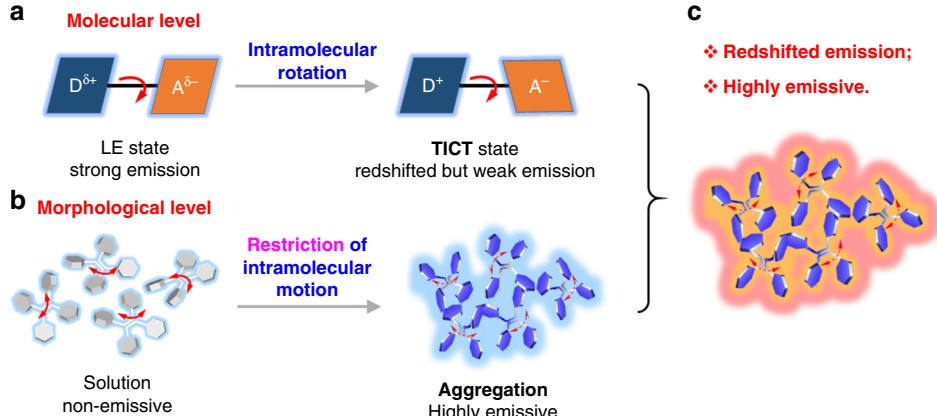

**Fig. 1 Schematic illustration of emission mechanism. a** The transition of the locally excited (LE) state to the TICT state by intramolecular rotation of the D-A units at the excited state (molecular level). **b** The mechanism of AIE through restriction of intramolecular motion (morphological level). **c** The combination of TICT and AIE in the aggregate state.

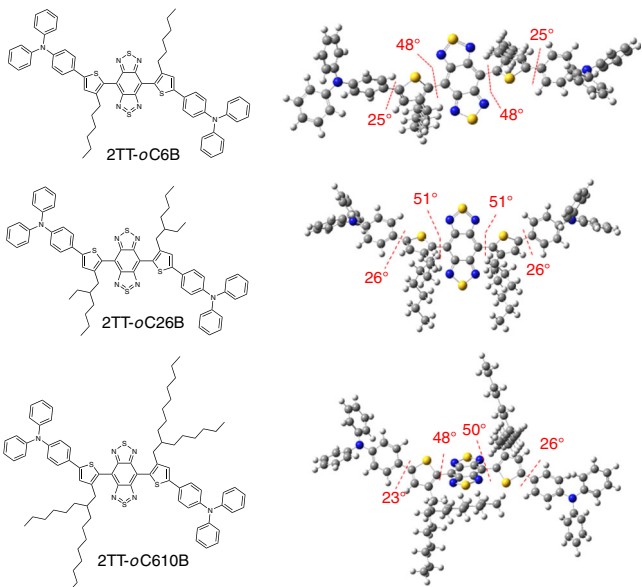

**Fig. 2** Chemical structures and optimized ground state ($S_0$) geometries of the molecules.

(Supplementary Figs. 9, 10). To inspect the fluorescence property of the three molecules in a higher morphology (aggregate state), we recorded their photoluminescence (PL) spectra in THF/$H_2O$ mixtures with different water fraction ($f_w$). The emission intensity of 2TT-oC26B decreased gradually with the addition of water into THF until $f_w = 40\%$, accompanied by a redshifted emission, demonstrating a significant solvation effect stemmed from TICT property (Fig. 3a and Supplementary Fig. 11). To further compare the TICT properties of the three dyes, the solvatochromic effect was studied. With an increase in solvent polarity, the absorption profiles were slightly changed, however, a remarkable decrease in emission intensity and redshifted wavelength were observed in their PL spectra (Supplementary Fig. 12). In order to better understand the photophysical property, a plot of Stokes shift ($\nu_{abs} - \nu_{em}$) versus the solvent polarity parameter ($\Delta f$) was attained based on the Lippert–Mataga equation (Supplementary Fig. 13). The relatively large slopes (>5000 cm$^{-1}$) are derived from the linear relationship, suggesting strong TICT properties[25].

To support the existence of TICT properties in aggregation, water was further added to the system with an increase of $f_w$ from 40 to 90%. The fluorescence intensity of 2TT-oC26B enhanced remarkably, owing to the RIM mechanism triggered by aggregate formation. Notably, the long-wavelength peak located at ~1030 nm indicated that the TICT property of 2TT-oC26B aggregates still remained. Similarly, 2TT-oC6B and 2TT-oC610B also exhibited "TICT + AIE" properties (Fig. 3b and Supplementary Fig. 14). Thus, the combination of the backbone (thiophene-BBTD-thiophene) distortion and donor (TPA) twisting benefited for the coexistence of the TICT and AIE effects, gathering the advantages of long-wavelength emission (TICT) and strong emission intensity (AIE) via manipulating intramolecular motions. To better study the emission in aggregate, we define $\alpha_{AIE}$ as the ratio of PL intensity at $f_w = 90\%$ to that of $f_w = 0$. Interestingly, 2TT-oC26B displayed a $\alpha_{AIE}$ of 10.8, which was higher than 2TT-oC610B (6.9) and 2TT-oC6B (6.0), suggesting a much stronger emission in aggregate. We reasoned that the differences in emission properties in aggregate have resulted from the different alkyl chains in these molecules. Compared with 2TT-oC6B, 2TT-oC26B with bulky 2-ethylhexyl unit possessed more twisted structures (Fig. 2), which helped impede detrimental intermolecular interactions in aggregate. Meanwhile, the 3D structure endowed molecules with molecular mobility. However, in 2TT-oC610B, the longer alkyl chain further facilitated the intramolecular motions of AIEgens aggregates to nonradiatively dissipate the excited energy[42].

To further decipher the fluorescence properties at morphological levels, we prepared the AIEgens into nanoparticles (AIE NPs) by nanoprecipitation method using biocompatible amphiphilic copolymers (DSPE-PEG$_{2000}$) as the doping matrix (Fig. 4a). On the one hand, the matrix endows AIEgens with excellent colloidal stability and desirable blood circulation time, attributed to the reduced immune recognition and minimized protein adsorption of surface PEG[47,48]. On the other hand, nanoparticles trigger the modulation of intramolecular motions to give bright and long-wavelength emission. As expected, the AIE NPs exhibited emission at the NIR-II region, similar to their solution-state profile. Noteworthily, their emission spectra extended to even 1600 nm (Fig. 3c), which is capable of NIR-IIb imaging. The relative longer emission of 2TT-oC6B (1034 nm) NPs than that of 2TT-oC26B (1031 nm) and 2TT-oC610B (1029 nm) is possible due to the better conjugation with the least distortion (Fig. 2). The QY of these AIE NPs were measured to be 11.5%, 9.1%, and 8.4% in the whole NIR-II region (1000–1600 nm) for 2TT-oC26B, 2TT-oC610B, and 2TT-oC6B, respectively, using IR-26 as reference (Fig. 3d, Supplementary Figs. 15–17 and

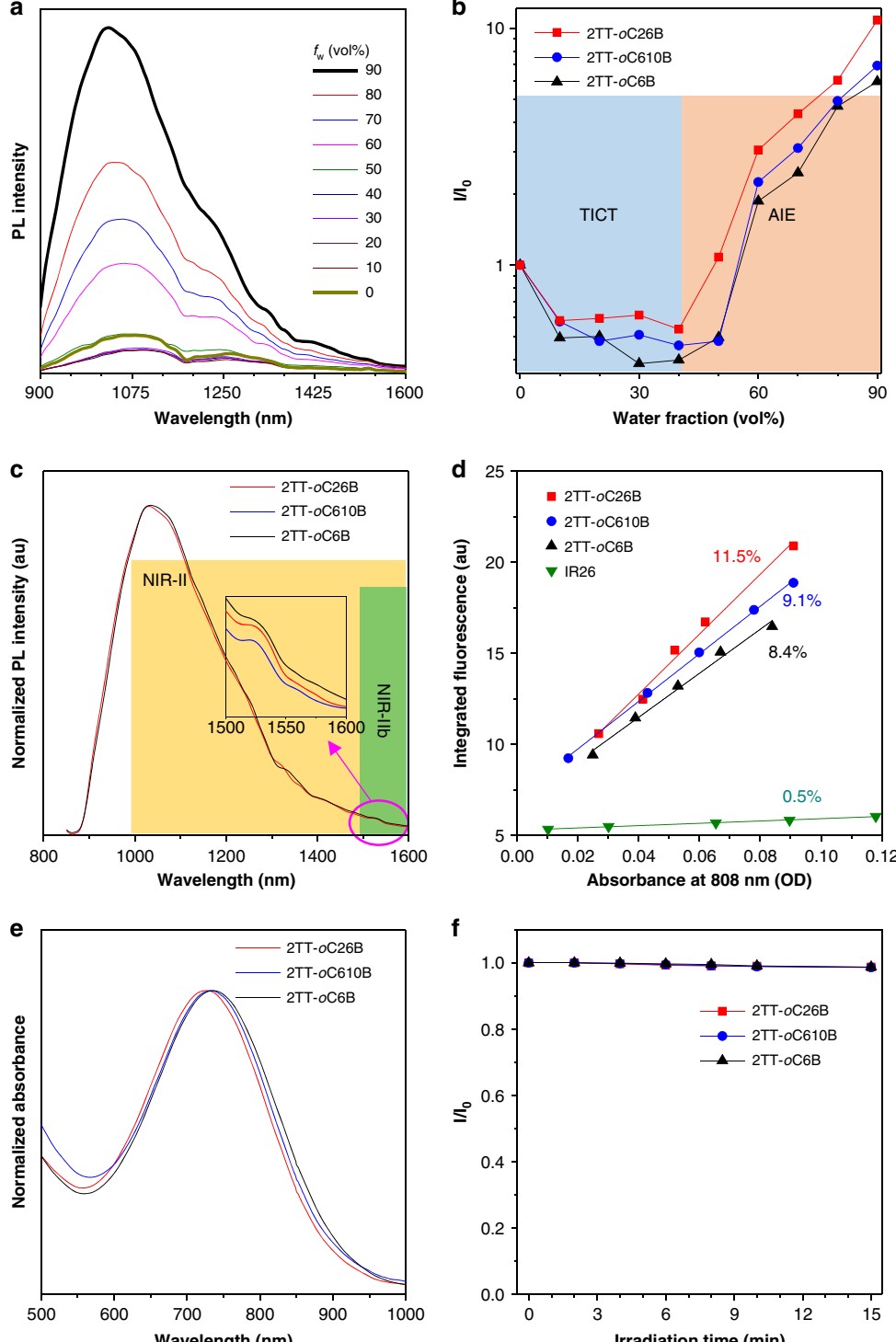

**Fig. 3 Photophysical properties of the compounds. a** PL spectra of 2TT-*o*C26B in THF/water mixtures with different water fractions ($f_w$). **b** Variation in PL intensity ($I/I_0$) of the three molecules with $f_w$, where $I$ and $I_0$ were the maximal PL intensity. **c** PL spectra of the nanoparticles. Inset: zoom-in emission spectra in the range of 1500–1600 nm. **d** The plots for the integrated fluorescence spectra of the three compounds nanoparticles (1000–1600 nm) and IR-26 (1050–1500 nm, QY = 0.5% in dichloroethane) at five different concentrations. **e** Absorption spectra of the nanoparticles. **f** The plot of absorption intensity ($A/A_0$) under continuous irradiation (110 mW/cm$^2$), where $A$ and $A_0$ were the maximal absorption intensity before and after laser irradiation, respectively. Source data are provided as a Source Data file.

Supplementary Table 1)[49]; while the QY in the NIR-IIb region (1500–1600 nm) was calculated to be 0.12%, 0.11% and 0.09%, respectively. The NIR-IIb QY of 2TT-*o*C26B NPs is significantly higher than the reported SWCNTs (QY = ~0.01%)[17]. The NIR-II fluorescence QY of 2TT-*o*C26B, 2TT-*o*C610B, and 2TT-*o*C6B in THF was much lower than that of aggregate (nanoparticle) state due to the non-radiative decay by strong molecular motions (Supplementary Fig. 18 and Supplementary Table 1). Notably, the absorption maxima of these AIE NPs located at 730 nm, which is useful for deep tissue excitation and avoiding photodamage to the

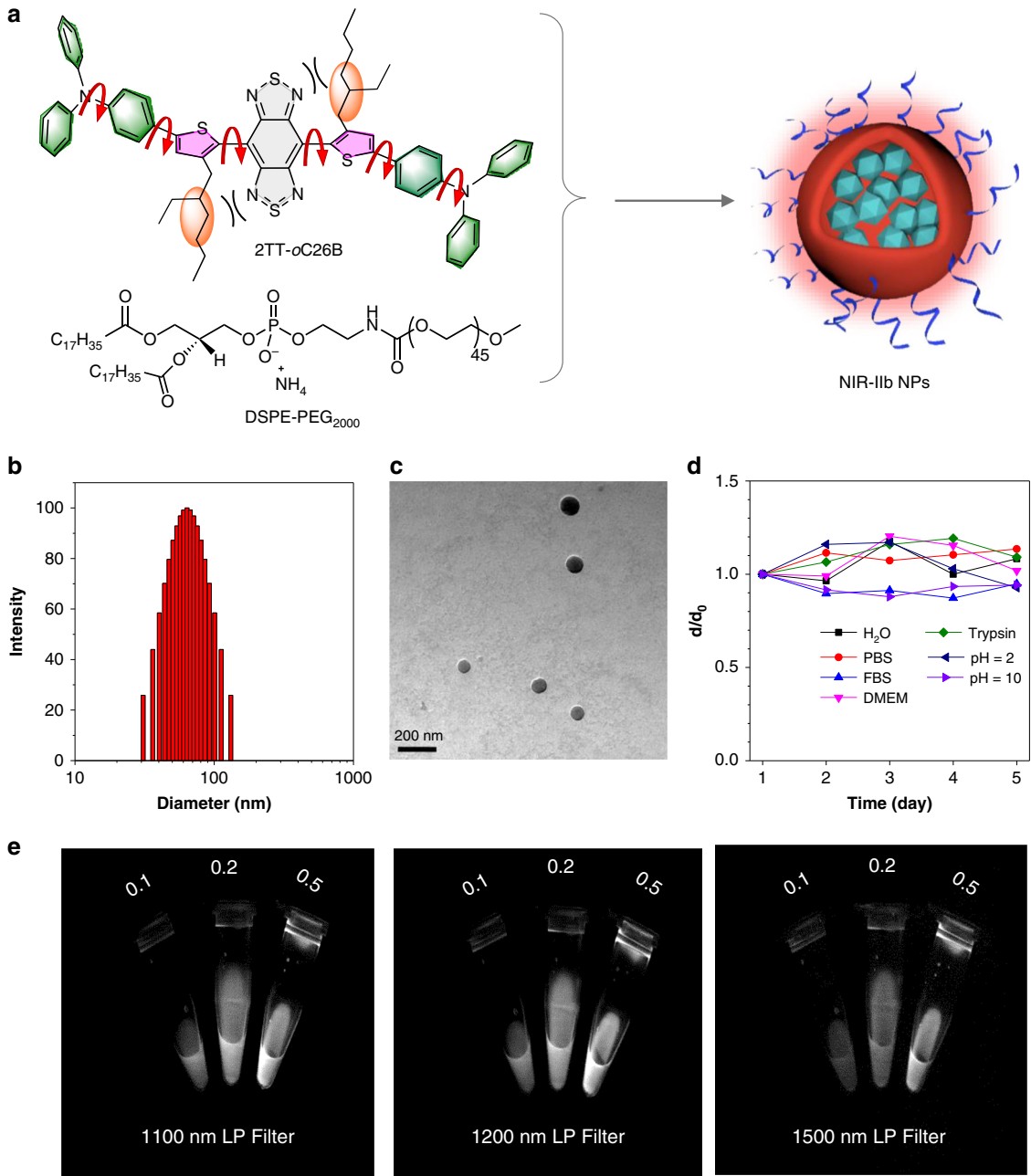

**Fig. 4 Characterization of 2TT-oC26B NPs. a** Schematic illustration of 2TT-oC26B NPs. **b** Representative DLS result and **c** TEM images. **d** Variation of diameter ratio ($d/d_0$) of 2TT-oC26B NPs in a different medium, where $d_0$ was the initial diameter. **e** Comparison of NIR-II signals under different LP filters at different concentrations (mg/mL).

organisms (Fig. 3e). Most importantly, these AIE NPs displayed good photostability under continuous laser irradiation (Fig. 3f). All these results are supportive that these AIE NPs could be utilized for NIR-IIb fluorescence imaging although their emission peaks are dominated at ~1030 nm.

**In vitro imaging**. Inspired by the redder emission and higher QY in NIR-IIb region of 2TT-oC26B, we evaluated its NIR-IIb imaging ability in vitro. The as-prepared 2TT-oC26B NPs exhibited good water dispersity with a homogeneous spherical structure, with diameters around 60 nm measured by dynamic light scattering (DLS) and transmission electron microscopy (TEM) (Fig. 4b, c). In addition, the 2TT-oC26B NPs showed excellent colloidal stability in various solutions, including water, Dulbecco's

modified eagle medium (DMEM), fetal bovine serum (FBS), phosphate buffer saline (PBS), trypsin and buffer solutions with pH = 2 and 10, and no precipitation occurred after 5 days (Fig. 4d and Supplementary Fig. 19). Moreover, the changes in absorption and emission for different solutions were negligible after storage for 5 days (Supplementary Fig. 20), which guarantees the efficiency of in vivo bioimaging. The images of 2TT-oC26B NPs with different concentrations (0.1, 0.2, 0.5 mg/mL) were recorded using three different long pass (LP) filters (1100, 1200, and 1500 nm) (Fig. 4e). When excited with a 793 nm laser, the NPs exhibited bright fluorescence in these three windows. Although the QY in the 1500–1600 nm region was only 0.12% (Supplementary Fig. 16), strong emission in NIR-IIb window could be observed. Taking the advantages of AIE effects, an increase in loading concentration of 2TT-oC26B increases the emission as

well. To further compare the bioimaging ability of 2TT-*o*C26B NPs in different NIR windows, the capillary tube filled with 2TT-*o*C26B NPs is immersed in a 1% intralipid solution at pointed phantom depths. As shown in Supplementary Fig. 21, even at 6 mm immersion depth, the clear tube boundary can be distinguished in the NIR-IIb region, but the tube is blurred and invisible in the NIR-I region. Although the brightest image is recorded in the NIR-II region owing to the highest QY, its SBR (1.8) and resolution according to the Gaussian-fitted full width at half maximum (FWHM = 0.86 cm) is significantly lower than those of in NIR-IIb (SBR = 3.1 and FWHM = 0.32 cm) based on the advantages of almost zero autofluorescence and much lower photon scattering. All these data suggest that the 2TT-*o*C26B NPs are suitable for NIR-IIb fluorescence imaging.

**Whole-body imaging**. Fluorescein angiography is a medical strategy by injecting a fluorescent probe into the bloodstream, which is of great importance for the circulatory system and disease diagnosis[50–52]. To further examine the advantages of NIR-IIb imaging, we intravenously injected 2TT-*o*C26B NPs into a mouse's bloodstream and its angiography was recorded by an InGaAs camera with different LP filters (1100, 1200, and 1500 nm). After intravenous injection of 2TT-*o*C26B NPs for 10 min, the whole vessel network of the mouse is clearly visualized (Fig. 5 and Supplementary Fig. 22). As compared with the traditional NIR-II imaging (1100 and 1200 nm LP filters), the NIR-IIb imaging exhibited superior resolution with an approximately transparent background (Fig. 5a–c). The cross-sectional intensity of similar capillaries (the red circle) was plotted to compare the SBR. The NIR-IIb window (1500 nm LP) had an SBR of 2.0, which was higher than that of the NIR-II windows (1.2 in 1200 nm LP and 1.1 in 1100 nm LP), demonstrating the advantages of

NIR-IIb imaging. Importantly, through measurement of FWHM of the selected region, the apparent widths of the similar vessels images with 1100, 1200, and 1500 nm LP were 0.58, 0.56 and 0.41 mm, respectively (Fig. 5d–f), suggesting that the NIR-IIb imaging offers the highest spatial resolution (Supplementary Fig. 22). In particular, the blood vessels closed to the liver cannot be clearly visualized by using 1100 and 1200 nm LP, while it was imaged clearly in 1500 nm LP. The high resolution and low background interference will provide more accurate diagnostic information on early diseases.

**Cerebral vasculature imaging**. Then, cerebral vasculature of Balb/c nude mouse through intact scalp and skull was further delineated in vivo after intravenous injection of 2TT-*o*C26B NPs. The cerebral vessel with a resolution of ~71.6 μm was distinctly observed (Fig. 6a–c), it is comparable with imaged by inorganic material[16,53]. To accurately detect the fine vessel structure, high-magnification through-skull microscopic vessel imaging of the brain was also conducted. As shown in Fig. 6d–f, the small vessel with an apparent width of only 10 μm can be apparently visualized. Such high resolution is achieved by organic molecules both in low and high-magnification imaging in NIR-IIb region[23]. Due to the significantly reduced autofluorescence and minimized photon scattering, 2TT-*o*C26B NPs exhibited high-performance NIR-IIb angiography with sharp resolution and high SBR, showing great advantages for in vivo imaging.

**Bowel imaging**. One of the limitations of fluorescence imaging is the penetration depth, hence it is difficult to "see-through" the body to monitor the inner soft tissues such as gastrointestinal (GI), whose disorder was associated with a multitude of diseases, including diabetes, thyroid disorders and colorectal cancer[54].

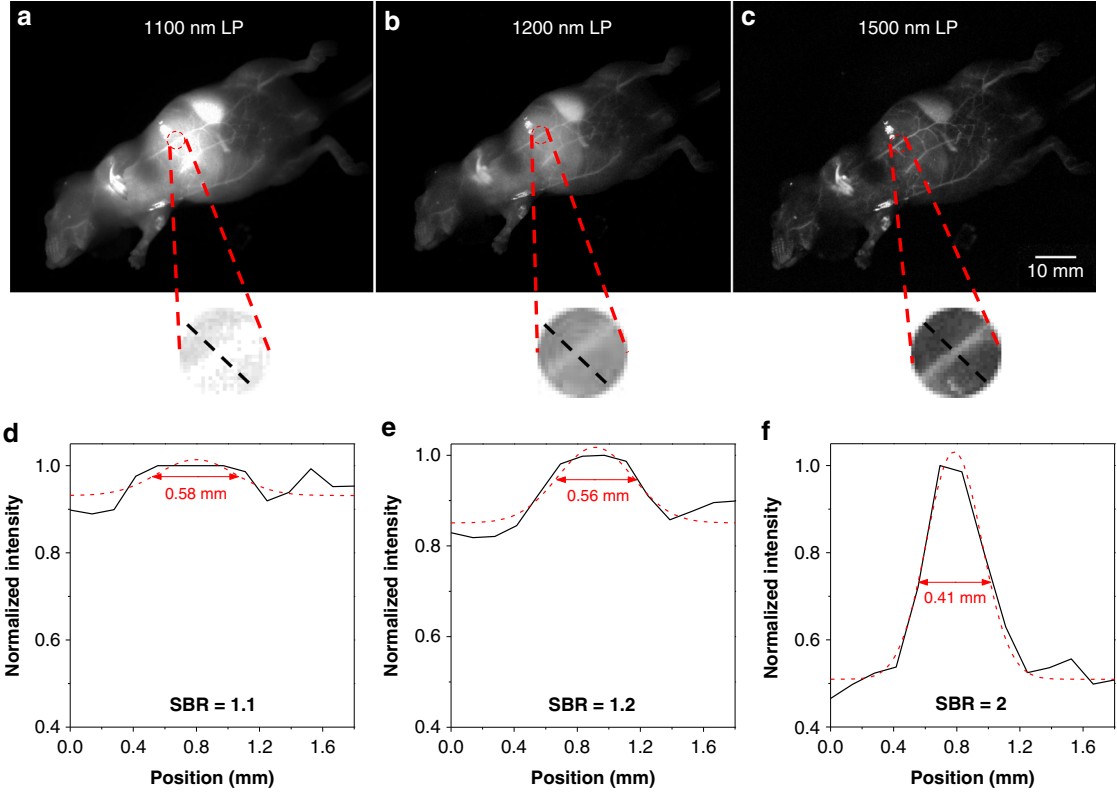

**Fig. 5 Comparison of NIR-II fluorescence signals for whole-body imaging of living mice in an area close to the liver under different LP filters treated with 2TT-*o*C26B NPs (200 μL, 0.8 mg/mL). a** 1100 nm LP, 5 ms, 37 mW/cm$^2$; **b** 1200 nm LP, 5 ms, 37 mW/cm$^2$; **c** 1500 nm LP, 150 ms, 75 mW/cm$^2$. **d–f** Corresponding cross-sectional fluorescence intensity profiles along black-dashed lines. Gaussian fits the profile are shown in the red line.

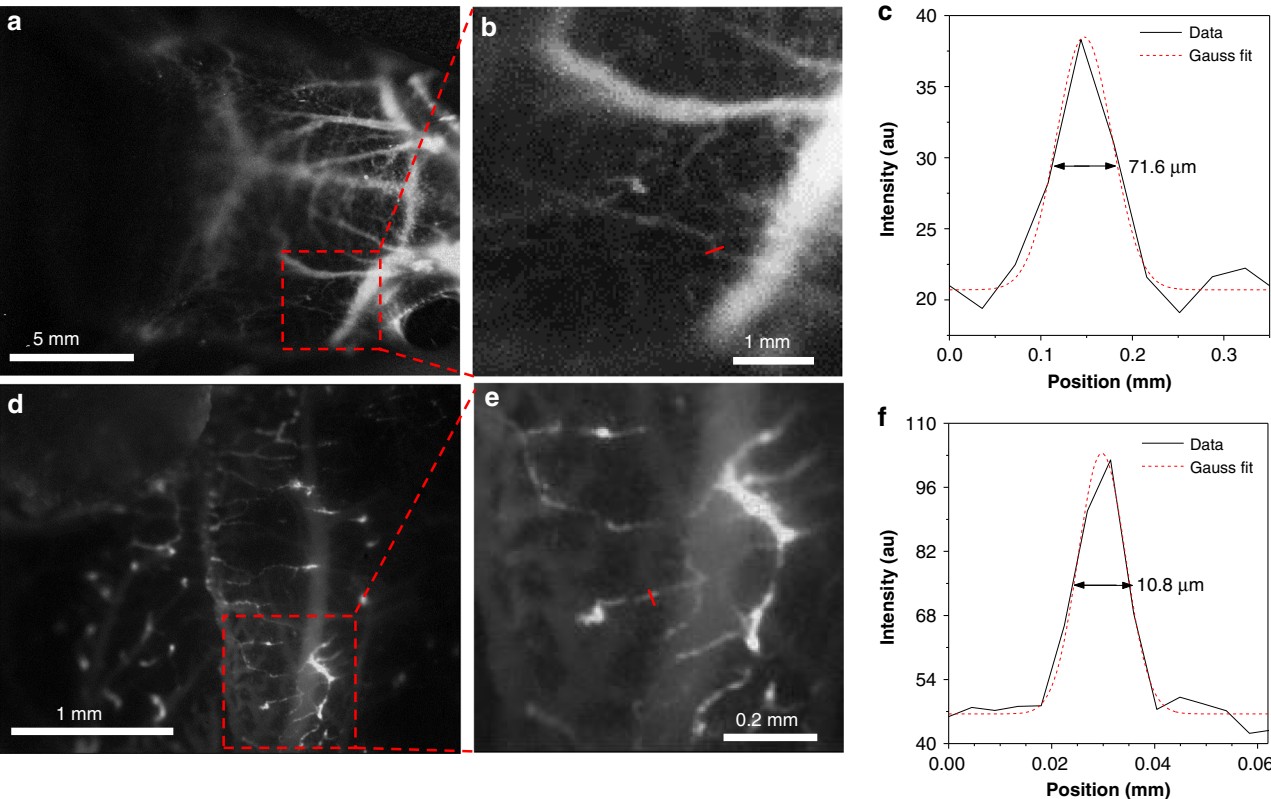

**Fig. 6 NIR-IIb fluorescence imaging of brain vasculature in living mice. a** NIR-IIb fluorescence image using a 50 mm fixed focal lens. **b** Region of interest from the red-dashed box in (**a**). **c** Cross-sectional fluorescence intensity profile along the red line shown in (**b**). **d** NIR-IIb fluorescence image using a scan lens (Thorlabs). **e** Region of interest from the red-dashed box in (**d**). **f** Cross-sectional fluorescence intensity profile along the red line shown in (**e**). Images were taken with a 793 nm laser excitation and the exposure time was 250 ms.

Although magnetic resonance imaging (MRI) and computed tomography (CT), have been commonly accepted to use in the clinic for the diagnosis of intestinal diseases[55,56], the limited spatial resolution, long imaging time, and the harmful radiation risk restricts monitoring the gut function[57–59]. NIR-IIb imaging, due to the superior temporal-spatial resolution, enables a platform for monitoring the bowel function in real-time. Hence, the intestinal structure was imaged with different LP filters (1100, 1200, and 1500 nm) at different time points after oral administration of 2TT-*o*C26B NPs (300 μL, 1 mg/mL). As shown in Fig. 7a, the ileum, cecum, colon, and rectum could be visualized after gavage at 0.5, 3, 5, and 6 h, respectively. Although the tissue could be detected with 1100 and 1200 nm LP, the images were blurry with low resolution. In contrast, clear resolution of tissue features was distinguishable with a negligible background using a 1500 nm LP filter in the NIR-IIb region. Importantly, even the intestinal tract located deep within at ~ 5 mm depth, the individual small bowel diverticula (~1 mm) was also clearly discriminated (Fig. 7). Longer imaging wavelengths enhanced the spatial resolution of fine structures of the intestinal tract by remarkably increasing the SBR (Fig. 7b, c and Supplementary Figs. 23, 24)[12]. At the same time, the cecum structure can also be clearly delineated at a reduced exposure time in 1500 nm LP (Supplementary Fig. 25). The contractile function of the intestine can be monitored clearly during the imaging procedure as shown in Supplementary Movie 1. Notably, even we use the rat as a model, its intestine structure can be observed with high clarity in the NIR-IIb region at a depth of ~8 mm while it is difficult to discriminate in NIR-I and NIR-II region (Supplementary Fig. 26). Subtle intestinal structures of mouse and rat can be monitored in such high resolution by using organic NIR-IIb probes. Finally,

after 24 h gavage feeding, 2TT-*o*C26B NPs were totally excreted from body in the form of feces (Supplementary Fig. 27), and were not entered the body via the intestine (Supplementary Fig. 28), which is beneficial for the development of oral GI diagnostic contrast agents[57]. Thus, 2TT-*o*C26B NPs could be a powerful platform for assessing diseases within deep tissues.

## Discussion

In conclusion, we demonstrated pure organic nanoparticles for high-quality NIR-IIb fluorescence imaging using the strategy of the combining of TICT with AIE. The key factor for molecular design is the union of backbone distortion with twisted molecular rotors for regulating of molecular motion in aggregates and the prevention of harmful intermolecular interactions. AIEgens are often propeller-like in shape and are, therefore, born with molecular mobility even in the aggregated state. On a molecular level, the twisted NIR-IIb emitters are favorable for intramolecular motion compared with planar molecules, resulting in the formation of TICT state. On a higher morphological level, molecular aggregates partially restrict the intramolecular motion of the molecules, giving a boost in fluorescence efficiency. Noteworthily, owing to the twisted 3D structures with multiple rotors, the AIEgens still remained intramolecularly mobile even in the aggregate state. Therefore, through structural modulations at molecular (TICT) and morphological levels (aggregation), organic NIR-II AIE-based nanoparticles with redshifted emission and high fluorescent QY were achieved simultaneously. The resultant 2TT-*o*C26B NPs displayed an emission spectrum extending to 1600 nm with a whole NIR-II (1000–1600 nm) QY of 11.5% and NIR-IIb (1500–1600 nm) QY of 0.12%. Meanwhile,

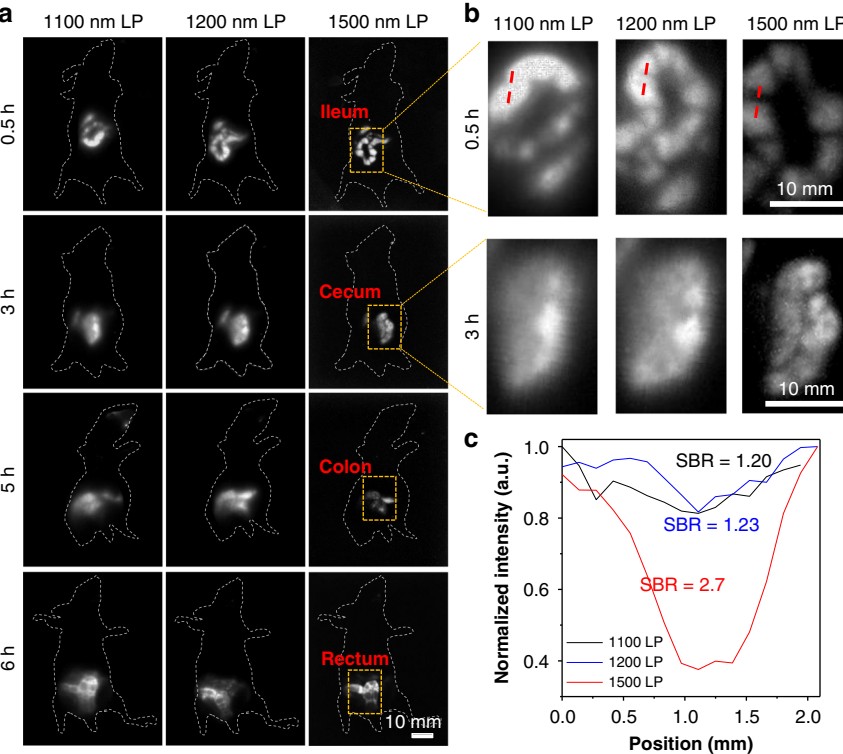

**Fig. 7 NIR-II fluorescence images of the intestinal tract. a** Real-time monitoring of intestinal peristalsis in living mice gavaged with the 2TT-oC26B NPs (300 μL, 1 mg/mL) using different LP filters (1100 nm LP, 8 ms, 37 mW/cm²; 1200 nm LP, 10 ms, 37 mW/cm²; 1500 nm LP, 75 ms, 110 mW/cm²) at various time points (0.5, 3, 5, 6 h). **b** Zoom-in images of the two yellow rectangle areas marked in (**a**) (0.5 and 3 h). Yellow-line dashed rectangles in the images at 5 and 6 h in (**a**) are zoomed-in and shown in Supplementary Fig. 24. **c** A cross-sectional fluorescence intensity profile along the red-dashed bar at the top of (**b**).

these organic NIR-II AIE NPs provide a platform for NIR-IIb fluorescence imaging of blood vasculature and intestinal tract with high quality. This study represents the pure organic nanoparticles for NIR-IIb imaging and will inspire further development of organic molecules with both ultralong emission wavelength and high brightness.

## Methods

**Fluorescence QY measurement.** The QY of the dyes was determined as follows using IR-26 as the reference (QY = 0.5%)[23,49]. IR-26 was diluted with 1,2-dichloroethane to a series of samples with their absorption intensity at 808 nm of ~0.02, ~0.04, ~0.06, ~0.08, ~0.1. The PL spectra were collected with an 880 nm LP filter to reject the excitation light (808 nm). Then the emission spectra were integrated in the 1050–1500-nm region. The same procedures were performed for the samples in water and THF too. The obtained emission integration was plotted against the absorption intensity and fitted into a linear relationship. The QY calculation Eq. (1) was as follows:

$$QY_{sample(wavelength)} = QY_{ref} \cdot \frac{S_{sample(wavelength)}}{S_{ref}} \cdot \left(\frac{n_{sample(wavelenth)}}{n_{ref}}\right)^2 \quad (1)$$

where $QY_{sample\ (wavelength)}$ is the QY of the nanoparticles in different region (wavelength 900–1600 nm, 1000–1600 nm for NIR-II, and 1500–1600 nm for NIR-IIb), $QY_{ref}$ is the QY of IR-26 (~0.5% in dichloroethane), $S_{sample}$ and $S_{ref}$ are the slopes obtained by linear fitting of the integrated emission spectra of the nanoparticles in different region (wavelength 900–1600 nm, 1000–1600 nm for NIR-II, 1500–1600 nm for NIR-IIb), and IR-26 (1050–1500 nm) against the absorbance at 808 nm, $n_{sample}$ and $n_{ref}$ are the refractive indices of $H_2O$ and dichloroethane, respectively.

**Fabrication of AIE NPs.** A mixture of AIEgen (1 mg), DSPE-PEG$_{2000}$ (2 mg), and THF (1 mL) was sonicated (12 W output, XL2000, Misonix Incorporated, NY) to obtain a clear solution. The mixture was quickly injected into water (9 mL), which was sonicated vigorously in water for 2 min. The mixture was stirred in fumehood overnight to remove the THF. AIE NPs was subjected to ultrafiltration (molecular weight cutoff of 100 kDa) at 3000 g for 30 min.

**Animal handling.** All the animal experiments were performed strictly in compliance with the requirements of the Zhejiang University Animal Study Committee. Nude mice and rats were obtained from the Laboratory Animal Center of Zhejiang University. They were housed at 24 °C with a 12 h light/dark cycle and fed with laboratory water and chow ad libitum.

**Image processing.** All images were processed using the same settings within a test for both controls and test samples.

**In vivo NIR-IIb imaging.** The whole-body blood vessel imaging and GI tract imaging were both carried out with a home-built NIR-IIb fluorescence imaging setup equipped with a 793 nm laser. The beam was coupled to a collimator and expanded by a lens, providing uniform irradiation on the imaging plane. A fixed focal lens was utilized to collect the signals and an InGaAs camera (TEKWIN SYSTEM, China, 900–1700 nm sensitive) was used to detect the fluorescence signals. For high-magnification microscopic imaging of the brain, a scan lens (Thorlabs) was equipped. The excitation beam was guided by the lens and the fluorescence signal was also recorded with it. 900 nm long pass (LP) and 1100 nm LP filters were placed between the camera and the imaging lens to get images above 1100 nm. In the same way, 900 nm LP and 1200 nm LP were used to obtained images above 1200 nm. 1500 nm LP was for imaging above 1500 nm. All these LP filters were from Thorlabs.

For in vivo whole-body imaging of mice, 200 μL 2TT-oC26B NPs (0.8 mg/mL) were injected intravenously into the blood vessels of the nude mice. The mice, before imaging, were anesthetized with pentobarbital. For in vivo cerebral vasculature imaging of mice, the dosage was 1.0 mg/mL (250 μL) via tail intravenous injection. For in vivo GI tract imaging of mice, 300 μL 2TT-oC26B NPs (1.0 mg/mL) and 3 mL 2TT-oC26B NPs (1.0 mg/mL) were perfused to the stomach of the nude mice and rats, respectively. For better handling, the mice were anesthetized with pentobarbital at various time post perfusion.

**Reporting summary.** Further information on research design is available in the Nature Research Reporting Summary linked to this article.

## Data availability

The source data underlying Fig. 3c, d and Supplementary Figs. 15a-d, 16d, 17 are provided as a Source Data file. All other data are available from the corresponding author.

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

## Acknowledgements

This project was financially supported by Zhejiang Provincial Natural Science Foundation of China LR17F050001, the National Science Foundation of China (21788102, 21805002, 61735016, and 61975172), the Research Grants Council of Hong Kong (A-HKUST605/16 and C6009-17G), the Innovation and Technology Commission (ITC-CNERC14SC01 and ITCPD/17-9), the Science and Technology Plan of Shenzhen (JCYJ20180507183832744), and the Ming Wai Lau Centre for Reparative Medicine Associate Member Programme.

## Author contributions

Y.L., J.Q., and B.Z.T. conceived and designed the experiments. Y.L., S.L., and S.T.H.W. performed the synthesis. H.Z. did the theoretical calculation. Z.C. and Y.L. performed the fluorescent imaging experiment. R.T.K.K. and J.W.Y.L. took part in the discussion and gave important suggestions. Y.L., Z.C., S.L., J.Q., and B.Z.T. co-wrote the paper.

## Competing interests

The authors declare no competing interests.
