## [Peer Review File · Nature Communications]

Reviewers' Comments:

Reviewer #1:

Remarks to the Author:

–This is a very nice contribution that is certainly worthy of publication in Nature Communications. The authors have delineated structural features in NIR dyes so that the emission wavelength falls into a region of the spectrum suitable for imaging through tissue and at the same time provide guidelines to increase the optical output through the restriction of emission quenching in the solid state. The resulting aggregates proved to be particularly promising for in vivo imaging, as illustrated in Figures 4 and 5.

Some additional comments follow:

The manuscript is well written and easy to follow. I am not too keen on the comparative discussion relative to other imaging materials, i.e. QDs, SWNT, etc. The work carried out by the authors stands on its own and is sufficient for publication in Nature Communications. Plus, it is not obvious that practitioners actually interested in translating to commercial application, i.e. in companies, have not obtained better performance, and have not disclosed the performance parameters and/or methods of preparation.

Line 64, it would be helpful to better describe what a “dark TICT state” refers to, presumably it refers to a non-emissive, or poorly emissive S1 excited state in which the molecular configuration is twisted relative to S0.

The logic for molecular design is very well articulated on Line 88.

On Line 153: the authors claim that the use of copolymer to stabilize chromophore suspensions lead to “desirable circulation time”. That is not obvious to me why it should be so. There should be some rationale, or a reference to previous work that demonstrates this to be the case.

What is the purpose of Figure S20 (labeled as Figure 20)? I am not an expert in intestinal peristalsis, imaging but the figure just does not seem very informative.

Line 170: should be “spherical” structure.

Reviewer #2:

Remarks to the Author:

This is one of a number of recent contributions to explore the use of the NIR-II wavelength range (1000 nm-1700 nm) in qualitative preclinical imaging. The authors have reported a NIR-IIb agent 2TT-oC26B with excitation at ~1000 nm and collection of emission light >1500 nm and test it in preclinical studies of the blood vessels and the intestinal tract. I believe that the authentic NIR-IIb dye (maximum emission wavelength >1500 nm) will be of great importance for in-vivo molecular imaging and image-guided diagnostics and surgery due to almost zero autofluorescence and much lower photon scattering. Overall, the manuscript is well organized and written, so I recommend publication after major revision.

1) The authors' group, Dai and coworkers have reported a similar concept and backbone structure of dyes that emit the fluorescence in the second near-infrared window (J. Am. Chem. Soc. 2019, 141, 13, 5359; Nat. Mater. 2016, 15, 235; Chem. Sci. 2016, 7, 6203). This study and experimental methods seem to be overlapped with the previous works except that the authors have designed three D-A typed AIEgens with emission extended to the NIR-IIb region. This may be an important development for AIE science because new and improved small-molecule NIR-IIb probes are realized and NIR-IIb imaging is urgently needed for future applications.

- 2) This contribution lacks the traceable measurements that are needed to draw the conclusions and speculations on the clinical applications that are made throughout the contribution. Comparison is made with preclinical tests of 1100 nm LP (8 ms, 37 mW/cm²), 1200 nm LP (10 ms, 37 mW/cm²), 1500 nm LP filter (75 ms, 110 mW/cm²), but excited at 808 with emission collected (non-optimally) at >1500 nm at different powers and time points. The exposure times of 70 ms are quite long compared to video rate NIR-II (8~10 ms) measurements routine. The authors make conclusions that are not substantiated by the results. The contribution is interesting, but not convincing as more rigorous testing is needed.
- 3) The authors demonstrate visualization of the intestinal tract located deep within at ~5 mm depth and the individual small bowel diverticula (~1 mm) - this is within the range of visible and NIR-I, NIR-II fluorophores and does not demonstrate the advantages of using NIR-II - the authors should use a larger animal model to show great depth of penetration.
- 4) The highest spatial resolution of the vessels for 1100, 1200, and 1500 nm LP was 0.58, 0.56 and 0.41 mm. Those were much lower than that of NIR-II agents in the literature (Biomaterials, 2018, 171, 153; Theranostics, 2019, 9, 3866, Nat. Commun. 2018, 9, 1171; 2014, 5, 4206, Nat. Med. 2012, 18, 1841) , Did the presented data have better results compared with other studies?
- 5) The compounds were poorly characterized. Based on the NMR data of these compounds, they were not pure. The NMR spectra data of the samples were not completely interpreted and assigned in the Supplementary information. The purity of these compounds must be insured, as this significantly affects the optical properties of the compounds. Thus, it is required to purify the compounds again and re-check all the related measurements including optical spectra, quantum yields et al.

Reviewer #3:

Remarks to the Author:

This manuscript reports the preparation of organic nanofluorophores for biological imaging in the NIR-IIb window. The nanofluorophores were formed by encapsulation of D-A typed AIEgens with alkyl thiophene bridge into amphiphilic copolymers, and displayed fluorescence emission tail extending to 1600 nm. The alkyl chain was tuned to enhance the quantum yield (QY), and the optimized nanofluorophore 2TT-oC26B was utilized for blood vasculature and intestinal tract imaging at NIR-IIb window, demonstrating high imaging quality. The importance of this work lies in the discovery of NIR-IIb emission based on D-A typed organic molecules for bioimaging. This is a remarkable result and has great general interest. It is strongly recommended for publication in the journal after addressing the following points:

- 1) The current discussion on the molecule design seems problematic and needs improvement. The authors mentioned "...twisted molecular rotors for enhanced molecular motion..." and "...molecular aggregation restrict the intramolecular motion...". These two points sound contradictory.
- 2) The authors claimed that the buckier alkyl chain could promote intramolecular motion, "which is conducive to the formation of dark TICT state". Then, how to explain that 2TT-oC6B with the least distortion shows the longest peak emission in Fig 2c?
- 3) Detailed optical data should be provided. For example, the PL spectra in Fig. 2a, S9 and Fig. S11 just cover up to 1300(1200) nm. Longer wavelength data in these conditions are helpful to understand the NIR-IIb emission in Fig 2c. The absorption coefficient and fluorescence quantum yields in organic solvents should also be given.
- 4) The QY calculation is confusing. Base on "Methods", the fluorescence signals from 900-1500 were used for QY calculation. However, the authors provided the QYs of 1000-1600 nm and 1500-1600 nm. The authors are recommended to provide the original data for these calculations. Fig 2d is strange as the integrated fluorescence for IR26 is negative, why? The authors may provide the original PL spectra with the appropriate unit and scale bar.
- 5) The way to calculate NIR-IIb QY based on the emission integration ratio may not be accurate. As the emission in NIR-IIb region is very weak and could be significantly affected by the background. Based on Fig 2d, the background signal is substantial. It is better to use the slope of

multiple points to get a more accurate NIR-IIb QY.

6) Line 24 "inorganic counterparts" should be "organic counterparts".

7) There is a recent paper on J-aggregate of cyanine dye for NIR-IIb imaging (DOI: 10.1021/jacs.9b10043). The authors may modify the claim on the "first" organic NIR-IIb fluorophore.

To Reviewer I:

This is a very nice contribution that is certainly worthy of publication in Nature Communications. The authors have delineated structural features in NIR dyes so that the emission wavelength falls into a region of the spectrum suitable for imaging through tissue and at the same time provide guidelines to increase the optical output through the restriction of emission quenching in the solid state. The resulting aggregates proved to be particularly promising for in vivo imaging, as illustrated in Figures 4 and 5.

1. The manuscript is well written and easy to follow. I am not too keen on the comparative discussion relative to other imaging materials, i.e. QDs, SWNT, etc. The work carried out by the authors stands on its own and is sufficient for publication in Nature Communications. Plus, it is not obvious that practitioners actually interested in translating to commercial application, i.e. in companies, have not obtained better performance, and have not disclosed the performance parameters and/or methods of preparation.

Reply: Thank you very much for your positive comment!

2. Line 64, it would be helpful to better describe what a “dark TICT state” refers to, presumably it refers to a non-emissive, or poorly emissive S1 excited state in which the molecular configuration is twisted relative to S0.

Reply: Thanks for your kind suggestion. The “dark TICT state” refers to the weakly emissive S1 excited state, as it efficiently quenches by various nonradiative processes. We have added the explanation in the revised manuscript on *Page 3 Paragraph 1 Line 1*.

3. The logic for molecular design is very well articulated on Line 88.

Reply: Thank you exceedingly for your positive comment.

4. On Line 153: the authors claim that the use of copolymer to stabilize chromophore suspensions lead to “desirable circulation time”. That is not obvious to me why it should be so.

There should be some rationale, or a reference to previous work that demonstrates this to be the case.

Reply: Thank you immensely for your valuable suggestion. We used the amphiphilic copolymer (DSPE-PEG₂₀₀₀) to stabilize the chromophore and hydrophilic PEG located on the surface of the nanoparticles. The surface PEG was beneficial for reducing immune recognition, minimizing protein adsorption and thereby increasing the circulation time in vivo. Previous works have demonstrated that the surface PEG segments were able to prolong the circulation time of DSPE-PEG nanoparticles in blood, which was highly desired in living animal studies (Angew. Chem. Int. Ed., 2013, 52: 10325-10329, Angew. Chem. Int. Ed., 2011, 50: 3430-3434, Chem. Soc. Rev., 2014, 43, 6570-6597). The explanation has been added in the revised manuscript *Page 7 Paragraph 2 Line 4*.

5. What is the purpose of Figure S20 (labeled as Figure 20)? I am not an expert in intestinal peristalsis, imaging but the figure just does not seem very informative.

Reply: Thank you very much for your kind suggestion. We experimented to confirm whether there are residual nanoparticles in the intestines. As shown in Supplementary Fig. 20, 2TT-*o*C26B NPs were excreted entirely from the body and there were no residual nanoparticles in the intestine. Although Supplementary Fig. 20 seems non-informative, it is crucial for biosafety and the development of new oral GI diagnostic contrast agent. Meanwhile, thank you for your reminder, the caption has been corrected (re-labeled with Supplementary Fig. 27) in revised Supporting Information.

6. Line 170: should be “spherical” structure.

Reply: Thank you very much for your kind reminder. The mistake has been corrected in the revised manuscript.

To Reviewer II:

This is one of a number of recent contributions to explore the use of the NIR-II wavelength range (1000 nm-1700 nm) in qualitative preclinical imaging. The authors have reported a NIR-IIb agent 2TT-oC26B with excitation at ~1000 nm and collection of emission light >1500 nm and test it in preclinical studies of the blood vessels and the intestinal tract. I believe that the authentic NIR-IIb dye (maximum emission wavelength >1500 nm) will be of great importance for in-vivo molecular imaging and image-guided diagnostics and surgery due to almost zero autofluorescence and much lower photon scattering. Overall, the manuscript is well organized and written, so I recommend publication after major revision.

1. The authors' group, Dai and coworkers have reported a similar concept and backbone structure of dyes that emit the fluorescence in the second near-infrared window (J. Am. Chem. Soc. 2019, 141, 13, 5359; Nat. Mater. 2016, 15, 235; Chem. Sci. 2016, 7, 6203). This study and experimental methods seem to be overlapped with the previous works except that the authors have designed three D-A typed AIEgens with emission extended to the NIR-IIb region. This may be an important development for AIE science because new and improved small-molecule NIR-IIb probes are realized and NIR-IIb imaging is urgently needed for future applications.

Reply: Thank you very much for your positive comment. The important works related to the NIR-II probes have been cited in the revised manuscript.

2. This contribution lacks the traceable measurements that are needed to draw the conclusions and speculations on the clinical applications that are made throughout the contribution. Comparison is made with preclinical tests of 1100 nm LP (8 ms, 37 mW/cm²), 1200 nm LP (10 ms, 37 mW/cm²), 1500 nm LP filter (75 ms, 110 mW/cm²), but excited at 808 with emission collected (non-optimally) at >1500 nm at different powers and time points. The exposure times of 70 ms are quite long compared to video rate NIR-II (8~10 ms) measurements routine. The authors make conclusions that are not substantiated by the results. The contribution is interesting, but not convincing as more rigorous testing is needed.

Reply: Thank you very much for your valuable comment. Since the QY of 2TT-oC26B nanoparticles in the NIR-IIb region (1500-1700 nm) is much lower than that of 1100-1700 nm and 1200-1700 nm, it is inevitable to choose higher power density and longer exposure time. It is

worth mentioning that the comparisons are conducted in their optimal conditions. Even if the power density or exposure time is increased, the resolution will not improve as much. Comparisons made with different power and exposure time were widely reported (Chem. Sci., 2019, 10, 1219, Biomaterials, 2018, 171, 153e163).

According to the review's suggestion, to make the manuscript more rigorous, we have reduced the exposure time at the cost of powder density (250 mW/cm^2) within safety limits (307 mW/cm^2) (Chem. Sci., 2019, 10, 1219). As shown in Fig. R1, the cecum structure can still be delineated clearly with high resolution. At the same time, the contractile function of the intestine can be monitored clearly during the imaging procedure as shown in Supplementary Movie 1. Although 40 ms and -40°C are the minimum video exposure time and lowest temperature that the camera can achieve, the real-time imaging of intestine with high clarity indicated the advantage of NIR-IIb imaging of 2TT-*o*C26B nanoparticles. Given the short imaging time and high resolution, it exhibited considerable promise for real-time gut function studies, including monitoring intestinal motility dysfunction and providing evaluation information for GI therapeutic agents in vivo. We have updated this in the revised manuscript (Page 12) and Supplementary Information (Supplementary Fig. 24).

Fig.R1 Real-time imaging of intestinal peristalsis in living mice gavaged with the 2TT-*o*C26B NPs ($300 \mu\text{L}$, 1 mg/mL) at NIR-IIb region (793 nm laser excitation 250 mW/cm^2 , exposure time: 40 ms).

3. The authors demonstrate visualization of the intestinal tract located deep within at $\sim 5 \text{ mm}$ depth and the individual small bowel diverticula ($\sim 1 \text{ mm}$) - this is within the range of visible and NIR-I, NIR-II fluorophores and does not demonstrate the advantages of using NIR-II - the authors should use a larger animal model to show great depth of penetration.

Reply: Thank you very much for your valuable suggestion. It is no doubt that imaging in visible, NIR-I, and NIR-II can visualize deep tissue to some extent. However, the limited resolution and SBR are their serious drawbacks in such depth. To further compare the bio-imaging ability of 2TT-*o*C26B NPs in different NIR windows, the capillary tube filled with 2TT-*o*C26B NPs is immersed in 1% Intralipid solution at pointed phantom depths. As shown in Fig. R2, even at 6 mm immersion depth, the clear tube boundary can be distinguished in the NIR-IIb region, but the tube is blurred and invisible in the NIR-I region. Although the brightest image is recorded in the NIR-II region due to the highest QY, its SBR (1.8) and resolution according to the Gaussian-fitted full width at half maximum (FWHM=0.86 cm) is significantly lower than those of in NIR-IIb (SBR=3.1 and FWHM=0.32 cm) based on the advantages of almost zero autofluorescence and much lower photon scattering. The results suggested that imaging in the NIR-IIb region exhibited the highest advantage than in the NIR-I and NIR-II region. We have updated this in the revised manuscript (Page 8) and Supplementary Information (Supplementary Fig. 20).

Fig. R2 (a) NIR fluorescence images of a capillary tube filled with 2TT-*o*C26B NPs immersed at depths of 2 mm (top) and 6 mm (bottom) in 1% Intralipid, recorded in NIR-I, NIR-II and NIR-IIb regions, respectively. (b, c) Cross-sectional fluorescence intensity profiles along yellow-dashed lines in the middle of capillary tubes. b: depth=2 mm, c: depth=6 mm.

According to the review's suggestion, a large animal (rat) was imaged after being gavage with the 2TT-oC26B NPs. Its intestine structure can be distinguished clearly in the NIR-IIb region at a depth ~ 8 mm while it is difficult to discriminate in the NIR-I and NIR-II region (Fig. R3). We have updated this in the revised manuscript (Page 12) and Supplementary Information (Supplementary Fig. 25).

Fig. R3 NIR fluorescence images of intestinal tract in living rat recorded in NIR-I, NIR-II and NIR-IIb regions. Rat was imaged at 5 hours after being gavage with the 2TT-oC26B NPs (1 mg/mL, 3 mL) under a 793 nm laser excitation with a power intensity less than 250 mW/cm^2 .

4. *The highest spatial resolution of the vessels for 1100, 1200, and 1500 nm LP was 0.58, 0.56 and 0.41 mm. Those were much lower than that of NIR-II agents in the literature (Biomaterials, 2018, 171, 153; Theranostics, 2019, 9, 3866, Nat. Commun. 2018, 9, 1171; 2014, 5, 4206, Nat. Med. 2012, 18, 1841), Did the presented data have better results compared with other studies?*

Reply: Thank you very much for your valuable suggestion, which is vital to improve the quality of our manuscript. In our previous work, we focused to demonstrate the imaging advantage of the NIR-IIb region, so we only conducted the whole-body imaging macroscopically but ignored monitoring the small vessel structure.

In the revised manuscript, cerebral vasculature of Balb/c nude mouse through the intact scalp and skull was observed in vivo after intravenous injection of 2TT-oC26B NPs (Fig. R4 a-c). The cerebral vessel with an apparent width (i.e., FWHM) of approximately $71.6 \mu\text{m}$ was distinctly recognized. Although the resolution is slightly lower than that ($43.65 \mu\text{m}$) in the reported work (Biomaterials, 2018, 171, 153), it is comparable and is the highest reported in the NIR-IIb region

by organic materials. Meanwhile, the limited resolution in our work is attributed to the InGaAs camera (TEKWIN), whose pixel size is 25 μm . However, in the aforementioned work (Biomaterials, 2018, 171, 153), the pixel size was as small as 15 μm .

To improve the spatial resolution, high-magnification through-skull microscopic vessel imaging of the brain was also conducted. As shown in Fig. R4 d-f, the small vessel with an apparent width of only 10 μm can be visualized. For the first time, such high resolution is achieved by organic molecules both in low and high-magnification microscopic imaging in the NIR-IIb region (J. Am. Chem. Soc. 2019, 141, 19221–19225). We have updated this in the revised manuscript (Page 10) and Fig. 5.

Fig. R4 NIR-IIb fluorescence imaging of brain vasculature in living mice. (a) NIR-IIb fluorescence image using a 50 mm fixed focal lens. (b) Region of interest from the red dashed box in (a). (c) Cross-sectional fluorescence intensity profile along the red line shown in (b). (d) NIR-IIb fluorescence image using a scan lens (Thorlabs). (e) Region of interest from the red dashed box in (d). (f) Cross-sectional fluorescence intensity profile along the red line shown in (e). Images were taken with a 793 nm laser excitation and the exposure time was 250 ms.

5. The compounds were poorly characterized. Based on the NMR data of these compounds, they were not pure. The NMR spectra data of the samples were not completely interpreted and assigned in the Supplementary information. The purity of these compounds must be insured, as this significantly affects the optical properties of the compounds. Thus, it is required to purify the compounds again and re-check all the related measurements including optical spectra, quantum yields et al.

Reply: Thanks for your critical comment. As you suggested, the compounds were purified, and their NMR spectra were provided, interpreted, and assigned in detail in Supplementary Fig. 1-6. After ensuring the purity of the compounds, all the optical spectra, quantum yields were remeasured and recalculated in the manuscript (Fig. 2, Supplementary Fig. 8-17 and Supplementary Table 1).

To Reviewer III:

This manuscript reports the preparation of organic nanofluorophores for biological imaging in the NIR-IIb window. The nanofluorophores were formed by encapsulation of D-A typed AIEgens with alkyl thiophene bridge into amphiphilic copolymers, and displayed fluorescence emission tail extending to 1600 nm. The alkyl chain was tuned to enhance the quantum yield (QY), and the optimized nanofluorophore 2TT-oC26B was utilized for blood vasculature and intestinal tract imaging at NIR-IIb window, demonstrating high imaging quality. The importance of this work lies in the discovery of NIR-IIb emission based on D-A typed organic molecules for bioimaging. This is a remarkable result and has great general interest. It is strongly recommended for publication in the journal after addressing the following points:

1. The current discussion on the molecule design seems problematic and needs improvement. The authors mentioned "...twisted molecular rotors for enhanced molecular motion..." and "...molecular aggregation restrict the intramolecular motion...". These two points sound contradictory.

Reply: Thanks for your kind reminder. However, these two points are not in conflict. Compared with traditional planar dyes, AIE luminogens (AIEgens) are often propeller-like in shape and, therefore, have molecular mobility even in the solid-state (ACS Materials Lett. 2019, 1, 425-431). In the aggregate state, the twisted NIR-IIb emitters are favorable for molecular motion compared to planar molecules, which in turn promotes formation of the TICT state. On the other hand, molecular aggregation partially restricts the intramolecular motions, giving strong fluorescence. It should be noted that AIEgens has molecular mobility in the aggregate state. Thereof, the TICT

state and high fluorescence efficiency can be achieved simultaneously in the aggregate state. Thus, these two points are not contradictory. As suggested, we have added the above explanations in the conclusion section of the revised manuscript.

2. The authors claimed that the buckier alkyl chain could promote intramolecular motion, “which is conducive to the formation of dark TICT state”. Then, how to explain that 2TT-oC6B with the least distortion shows the longest peak emission in Fig 2c?

Reply: Thanks for your kind reminder. Indeed, 2TT-oC6B with the least distortion shows the most prolonged peak emission in Fig 2c. This phenomenon is attributed to the better conjugation of 2TT-oC6B with the least distortion. Compared to the other two molecules with more considerable distortion, 2TT-oC6B displays inferior fluorescent quantum yield in the aggregate state due to the intermolecular interactions. Despite increasing conjugation length being more efficient to red-shift the emission, the emission is easily quenched owing to the strong intermolecular interactions. We have added the above explanations in the revised manuscript on *Page 7 Paragraph 2 Line 9*.

3. Detailed optical data should be provided. For example, the PL spectra in Fig. 2a, S9 and Fig. S11 just cover up to 1300 (1200) nm. Longer wavelength data in these conditions are helpful to understand the NIR-IIb emission in Fig 2c. The absorption coefficient and fluorescence quantum yields in organic solvents should also be given.

Reply: Thank you for your kind suggestion. According to your comments, all the PL in Fig. 2a, S9, S11 spectra were remeasured extended to 1600 nm in the revised manuscript. The molar absorption coefficient of 2TT-oC6B, 2TT-oC26B and 2TT-oC610B molecule in THF is 2.49×10^4 , 2.25×10^4 , 2.13×10^4 L mol⁻¹ cm⁻¹, respectively. The fluorescence quantum yields in organic solvents were measured and calculated (Supplementary Fig.17 and Supplementary Table 1).

4. The QY calculation in confusing. Base on “Methods”, the fluorescence signals from 900-1500 were used for QY calculation. However, the authors provided the QYs of 1000-1600 nm and 1500-1600 nm. The authors are recommended to provide the original data for these calculations. Fig 2d is strange as the integrated fluorescence for IR26 is negative, why? The authors may provide the original PL spectra with the appropriate unit and scale bar.

Reply: Thanks for your suggestion. We are sorry for miswriting 900-1500 nm as 1000-1600 in the manuscript. In our previous version, we calculated the whole NIR-II QY (900-1500) as the NIR-II QY according to the reference (Nat. Commun., 2014, 5, 4206). The QYs of 1500-1600 nm were calculated according to the integration area ratio ($QY_{1500-1600\text{ nm}} = QY_{\text{NIR-II}} * A_{1500-1600\text{ nm}} / A_{900-1500}$). As you suggested, the original data used for calculation the QY was provided in Fig. R5, and the source data was provided as a Source Data file.

Fig. R5 Quantum yield measurement. NIR-II fluorescence emission of nanoparticles (a, 2TT-*o*C26B; b, 2TT-*o*C610B; c, 2TT-*o*C6B) and (d) IR26 (dichloroethane) with different concentrations.

The negative integrated fluorescence of IR26 is because we used the same conditions to measure the PL spectra of the sample. Hence the negative noise of IR26 is evidently due to the detector of the PL machine. Thus, the integrated fluorescence of IR26 is negative. However, the negative, integral area does not affect the slope calculation, so we did not calibrate the curve in our

previous result. The original data with scale bar was provided in Fig. R1d, and source data was provided as a Source Data file.

To reduce errors, we used a new PL instrument (Horiba iHR 320 with InGaAs detector) whose baseline was calibrated well. The QYs from 1000-1600 nm as NIR-II QY and 1500-1600 nm as NIR-IIb QY were all remeasured using the slope of multiple points (Fig. 2d, Supplementary Fig. 15-16). The value of QYs was concluded in Supplementary Table 1. The original data was provided as a Source Data file.

5. The way to calculate NIR-IIb QY based on the emission integration ratio may not be accurate. As the emission in NIR-IIb region is very weak and could be significantly affected by the background. Based on Fig 2d, the background signal is substantial. It is better to use the slope of multiple points to get a more accurate NIR-IIb QY.

Reply: Thanks for your kind suggestion. We have prepared the samples and measured its PL spectra. According to your advice, to get a more accurate result, the QYs of 1000-1600 nm and 1500-1600 nm were calculated using the slope of multiple points (Fig. 2d, Supplementary Fig. 14-16). The value of QYs was concluded in Supplementary Table 1.

6. Line 24 “inorganic counterparts” should be “organic counterparts”.

Reply: Thanks for your kind reminder. The mistake has been corrected in the revised manuscript.

7. There is a recent paper on J-aggregate of cyanine dye for NIR-IIb imaging (DOI: 10.1021/jacs.9b10043). The authors may modify the claim on the “first” organic NIR-IIb fluorophore.

Reply: Thanks for your suggestion. This excellent paper attracted our attention when the manuscript was under consideration. According to your suggestion, we have revised the manuscript and cited the paper.

Reviewers' Comments:

Reviewer #1:

Remarks to the Author:

The authors have done a proper effort in revising the manuscript so that it meets the standards of Nature Communications.

Reviewer #2:

Remarks to the Author:

Comments were well addressed and the revised version is suitable for publishing in Nat. Commun.

Reviewer #3:

Remarks to the Author:

The authors have made a good revision on this manuscript, and substantial data are added. We suggest the acceptance for publication after addressing the minor issues below.

- 1) On Line 271, "molecular motion compared to planar...", does it mean intramolecular, intermolecular, or both?
- 2) Based on the data in Supplementary Table 1, the authors did not count the refractive index difference in QY calculation.
- 3) On Line 222-223, it is not suitable to compare the injection dose with other NIR-IIb fluorophores if afforded SBR values are very different.
- 4) On Line 181 "which overcome the limitation of concentration available for in vivo application". In my opinion, the limitation of injected concentration is not mainly due to the concentration quenching in most cases.
- 5) Based on Supplementary Fig 15 and 17, the spectra beyond 1500 nm are noisy. The authors may add the spectrum of blank solution to show the background. What is the peak at 1530 nm?

To Reviewer #1:

The authors have done a proper effort in revising the manuscript so that it meets the standards of Nature Communications.

Reply: Thank you very much for your positive comment!

To Reviewer #2:

Comments were well addressed and the revised version is suitable for publishing in Nat. Commun.

Reply: Thank you very much for your positive comment!

To Reviewer #3:

The authors have made a good revision on this manuscript, and substantial data are added. We suggest the acceptance for publication after addressing the minor issues below.

1) On Line 271, “molecular motion compared to planar...”, does it mean intramolecular, intermolecular, or both?

Reply: Thanks for your kind reminder. It should be intramolecular motion, and we revised it in the updated manuscript. Twisted molecules full of molecular rotors are favorable for intramolecular motions compared to planar ones.

2) Based on the data in Supplementary Table 1, the authors did not count the refractive index difference in QY calculation.

Reply: Thanks for your kind reminder. For the QY in nanoparticles, we count the refractive index of both 1,2-dichloroethane and water. For the QY in THF, indeed, we ignored the refractive index of THF. Therefore, we have recalculated the fluorescence QY of the samples in THF by counting the refractive index of THF. We have updated the data in Supplementary Table 1.

3) *On Line 222-223, it is not suitable to compare the injection dose with other NIR-IIb fluorophores if afforded SBR values are very different.*

Reply: Thanks for your valuable suggestion. We have removed the description about the comparison of injection dose with other NIR-IIb fluorophores from the manuscript.

4) *On Line 181 “which overcome the limitation of concentration available for in vivo application”. In my opinion, the limitation of injected concentration is not mainly due to the concentration quenching in most cases.*

Reply: Thanks for your kind reminder. Indeed, the limitation of concentration is not determined by the concentration quenching in most cases. To avoid misunderstandings, we have removed the description of “*which overcome the limitation of concentration available for in vivo application*” from the revised manuscript.

5) *Based on Supplementary Fig 15 and 17, the spectra beyond 1500 nm are noisy. The authors may add the spectrum of blank solution to show the background. What is the peak at 1530 nm?*

Reply: Thanks for your valuable advice. Followed your suggestion, we have measured the PL spectra of blank solution (water and THF), as shown in Supplementary Fig S15 and S17. The results showed that the emission peak at 1530 nm existed in the blank solution as well, indicating that it was caused by the PL machine itself. According to the QY equation, the integration of the peak from all curve will affect the intercept rather than the slope. Thus, the QY of the samples will not be influenced by the peak at 1530 nm.

Reviewers' Comments:

Reviewer #3:

Remarks to the Author:

The authors have addressed all the issues. Thus, I recommend its publication in the journal.